# Increasing reproducibility, robustness, and generalizability of biomarker selection from meta-analysis using Bayesian methodology

**Laurynas Kalesinskas**[1,2,3], **Sanjana Gupta**[1,2], **Purvesh Khatri**[1,2]*

**1** Institute for Immunity, Transplantation and Infection, School of Medicine, Stanford University, Stanford, California, United States of America, **2** Center for Biomedical Informatics Research, Department of Medicine, Stanford University, Stanford, California, United States of America, **3** Department of Biomedical Data Science, School of Medicine, Stanford University, Stanford, California, United States of America

* pkhatri@stanford.edu

**Data Availability Statement:** The bayesMetaIntegrator R package is publicly available for use at https://github.com/Khatri-Lab/

## Abstract

A major limitation of gene expression biomarker studies is that they are not reproducible as they simply do not generalize to larger, real-world, heterogeneous populations. Frequentist multi-cohort gene expression meta-analysis has been frequently used as a solution to this problem to identify biomarkers that are truly differentially expressed. However, the frequentist meta-analysis framework has its limitations–it needs at least 4–5 datasets with hundreds of samples, is prone to confounding from outliers and relies on multiple-hypothesis corrected p-values. To address these shortcomings, we have created a Bayesian meta-analysis framework for the analysis of gene expression data. Using real-world data from three different diseases, we show that the Bayesian method is more robust to outliers, creates more informative estimates of between-study heterogeneity, reduces the number of false positive and false negative biomarkers and selects more generalizable biomarkers with less data. We have compared the Bayesian framework to a previously published frequentist framework and have developed a publicly available R package for use.

## Author summary

There has long been a reproducibility crisis in medical research–driven by small, single-cohort studies with low-to-moderate statistical power. One of the reasons for this lack of generalizability is not accounting for heterogeneity representative of the real-world patient population. To address this issue, researchers have turned to meta-analysis–which allows for researchers to combine data from across previously published studies to generate an overall estimate of an effect, which has been used with gene expression data to create diagnostic and prognostic markers of disease. However, traditional meta-analysis techniques have limitations–they need at least 4–5 datasets with hundreds of samples and are prone to confounding from outliers in datasets. In this study, we create a new framework for gene expression meta-analysis using Bayesian statistics and show that it is more robust to outliers, creates more informative estimates of heterogeneity, reduces the

bayesMetaIntegrator. All data used in this study is publicly available – identifiers of which are found in S1–S3 Tables.]

**Funding:** PK is funded in part by the Bill and Melinda Gates Foundation (OPP1113682); the National Institute of Allergy and Infectious Diseases (NIAID) grants 1U19AI109662 and U19AI057229; Department of Defense contracts W81XWH-18-1-0253 and W81XWH1910235; and the Ralph & Marian Falk Medical Research Trust. The funders had no role in study design, data collection and analysis, decision to publish, or preparation of the manuscript.

**Competing interests:** The authors have declared that no competing interests exist.

amount of data required, and reduces the number of false positive and false negative biomarkers. We have compared the Bayesian framework to a previously published framework and have developed a publicly available R package for use.

## Introduction

With the advent of high-throughput transcriptomics, researchers have been able to profile gene expression in millions of samples at low costs, opening many new avenues of research. However, single-cohort studies with low-to-moderate statistical power are one of the four horsemen of irreproducibility in biomedical research [1]. Recent studies have estimated that only 10–25% of biomedical studies are reproducible [2–4]. This is especially a problem in biomarker studies, where differentially expressed genes (DEGs) between subjects with disease of interest and controls rarely generalize to the real-world patient populations. One of the reasons for this lack of generalizability is not accounting for heterogeneity representative of the real-world patient population.

Broadly, there are three sources of heterogeneity in the real-world patient population: biological (age, sex, tissue, cell type), clinical (treatment, disease duration, comorbidities), and technical (experimental protocol, batch effects). Traditionally, in a single cohort analysis, these sources of heterogeneity reduce the statistical power, requiring a large number of samples. Therefore, single cohort studies strive to increase statistical power by limiting heterogeneity as much as possible. However, this reduction in heterogeneity in single cohort studies leads to reduced generalizability to heterogeneous, real-world patient populations. We have repeatedly shown that leveraging heterogeneity across independent cohorts using a frequentist meta-analysis approach can identify robust disease signatures that are diagnostic and prognostic, and have been translated into a point-of-care test for clinical use [5,6]. We have previously identified best practices for gene expression meta-analyses, such as the number of studies and samples needed [7].

Despite its repeated success in identifying robust disease signatures, the frequentist approach has its limitations. First, previous work has shown that approximately 4–5 datasets with about 250 samples are needed to perform a successful frequentist meta-analysis [7]. However, several diseases simply do not have enough samples or datasets publicly available for successful integration using this guideline. Second, the statistic used to estimate and summarize effect sizes (e.g., Cohen's $d$, Hedge's $g$) can be susceptible to outlier samples within a subset of studies, resulting in misleading effect size estimates. Finally, frequentist approaches rely on multiple hypotheses corrected p-values, which are shown to be substantially underestimated [7]. Bayesian meta-analysis approaches have the potential to overcome these limitations. For example, Bayesian estimation has previously been shown to be more outlier resistant than traditional hypothesis testing [8]. Importantly, unlike frequentist meta-analysis, adjusting for multiple comparisons is not required for Bayesian approaches and yields more efficient and reliable estimates of effect [9].

We compared a frequentist approach with a new framework utilizing Bayesian approximation supersedes the t-test (BEST) for the meta-analysis of transcriptome using multiple independent datasets from humans with different diseases [8]. Here, we show that using this Bayesian approach, we are able to: 1) select more generalizable and robust biomarkers with fewer datasets, 2) be robust to outliers, 3) create better estimates of between study heterogeneity for biomarker selection, and 4) reduce the number of false positives and false negative genes for classification. This framework has also been developed into an R package,

bayesMetaIntegrator, that is publicly available for use (https://github.com/Khatri-Lab/bayesMetaIntegrator).

## Results

### Bayesian meta-analysis is resistant to outliers and provides a better estimate of heterogeneity in gene expression meta-analysis

We investigated whether the two meta-analysis approaches, Bayesian (**Fig 1**) and frequentist, identified the same or different set of genes using four publicly available asthma bronchial epithelial cell gene expression datasets—differentiating samples from asthma patients and healthy controls (**S1 Table**) [10–12]. Although the summary effect sizes were highly correlated between approaches (r = 0.94, p<2.2e-16; **Fig 2A**), the Bayesian approach consistently estimated higher between-dataset heterogeneity, $\tau^2$, than the frequentist approach (**Fig 2B**). While the frequentist approach found a large number of genes (26%) with no between-dataset heterogeneity, the Bayesian approach did not assign near-zero heterogeneity ($\tau^2 < 0.01$) to any genes. This difference in $\tau^2$ between the two approaches is due to how it is estimated. In a frequentist meta-analysis, high within-study heterogeneity leads to wider confidence intervals, which in turn drowns out the between-study heterogeneity. In contrast, Bayesian meta-analysis uses a probabilistic distribution to represent $\tau^2$ instead of a confidence interval, resulting in more conservative estimates of heterogeneity. The range of between-datasets heterogeneity was substantially higher for the Bayesian approach compared to the frequentist approach, further suggesting a Bayesian approach is more conservative than a frequentist approach.

Interestingly, despite the high correlation between summary effect sizes, within-dataset effect size correlations ranged widely from 0.87 to 0.95. Closer examination found that both approaches differed substantially in effect size estimates for a subset of genes. For example, in GSE64913, a subset of genes has an effect size of 0 when using the Bayesian approach, but a non-zero effect size when using a frequentist meta-analysis (**Fig 2B**). Therefore, we investigated whether these differences in effect size estimates and between-dataset heterogeneity will lead to identification of different set of differentially expressed genes (DEGs).

We used false discovery rate (FDR) for the frequentist approach and Bayesian probability for the Bayesian approach as the measures of statistical significance. Low correlation between both measures (r = 0.37, p<2.2e-16) suggested they identified different sets of DEGs. As the number of significant genes by either approach increased, the number of genes identified as

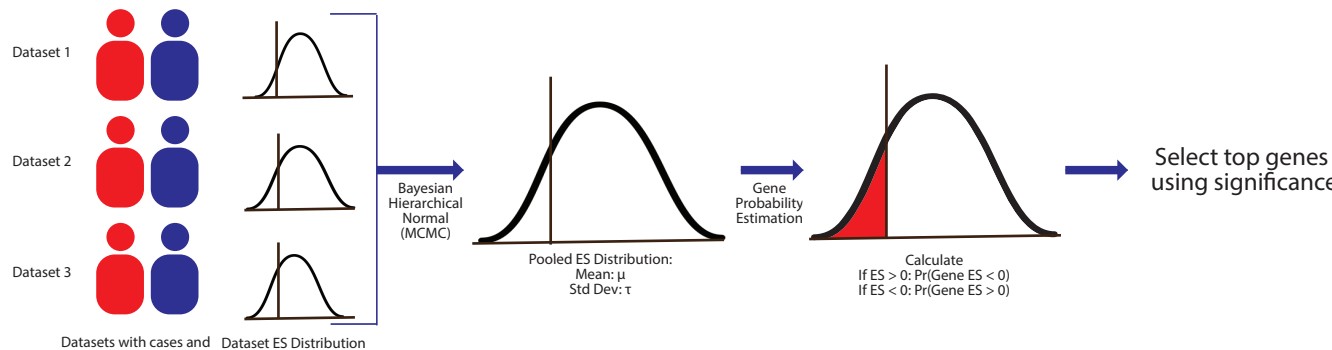

**Fig 1. Bayesian gene expression schematic.** We first use the BEST framework to estimate the posterior distribution of effect size between cases and controls for each gene in each dataset. Then we combine the distributions from independent studies using a gaussian hierarchical model, estimating both the pooled effect size and between-study heterogeneity in the process. Following, we estimate the probability of a gene being upregulated or downregulated based on the pooled effect size distribution.

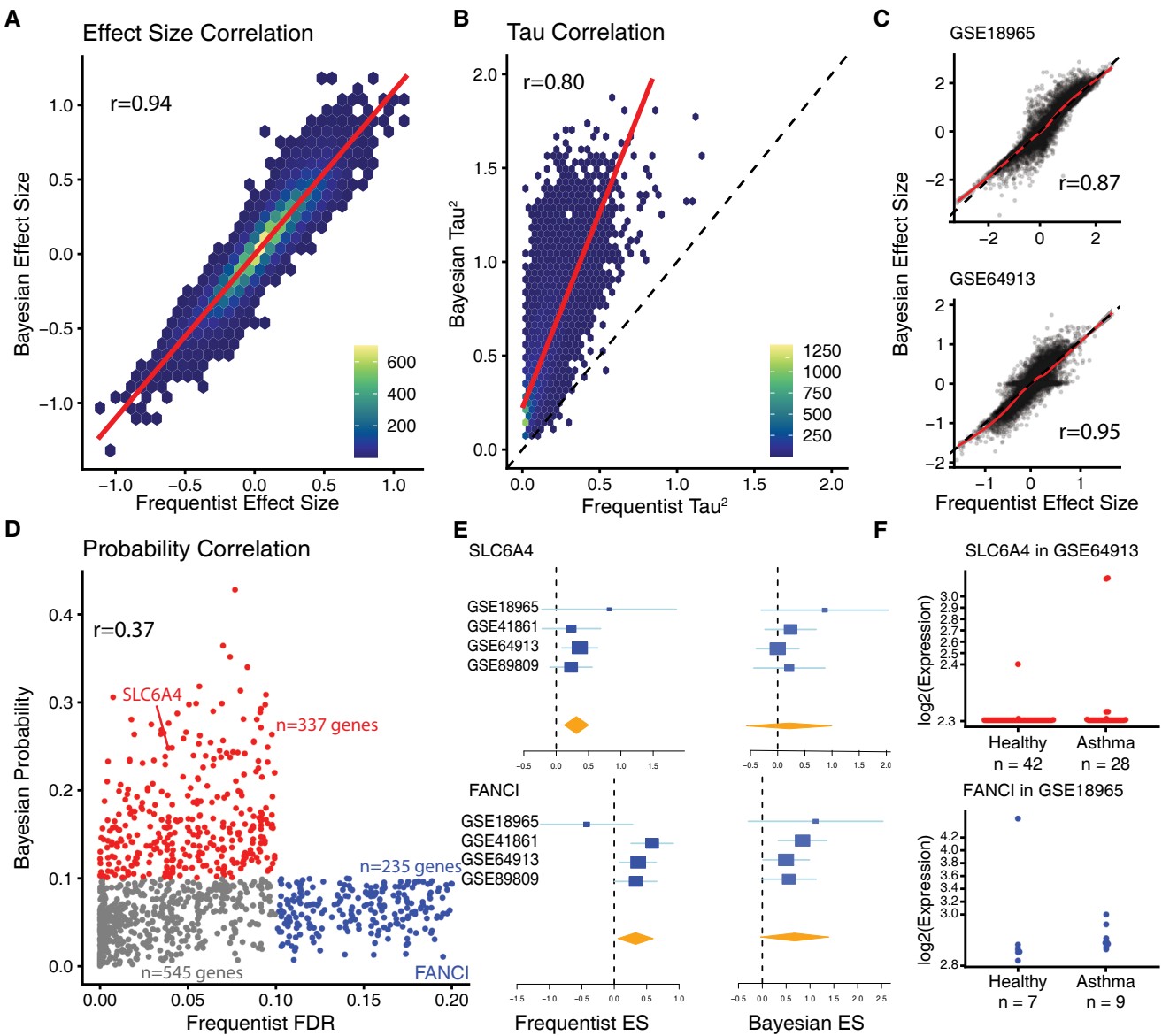

**Fig 2. Asthma Meta-Analysis Comparison.** (A) Comparison of pooled effect sizes by the frequentist and Bayesian meta-analysis for all genes. Pearson correlation (r) = 0.94. (B) Comparison of between-dataset heterogeneity, $\tau^2$, by the frequentist and Bayesian meta-analysis for all genes. Pearson correlation (r) = 0.80. (C) Comparison of within-dataset effect size estimates by the frequentist and Bayesian meta-analysis for all genes. (D) Comparison between false discovery rate from the frequentist meta-analysis and probabilities from the Bayesian meta-analysis. (E) The x axes represent standardized mean difference between patients with asthma and healthy controls, computed as Hedges' $g$, in $\log_2$ scale. The size of the blue rectangles is proportional to the standard error of mean difference in the study. Whiskers represent the 95% confidence interval. The orange diamonds represent overall, combined mean difference (summary effect size) for a given gene. Width of the orange diamonds represents the 95% confidence interval of overall mean difference. (F) Expression of SLC6A4 in GSE64913 and FANCI in GSE18965 in patients with asthma and healthy controls.

significant by both methods also increased (**S3 Fig**). Out of 1,117 DEGs that were statistically significant by either approach, 545 genes (48.8%) were statistically significant by both approaches (FDR < 10% and Bayesian probability < 0.1). The remaining 572 genes were significant when using either the frequentist approach (235 genes, 21%) or the Bayesian approach (337 genes, 30.2%) (**Fig 2D**).

Importantly, for genes that were significant by either approach but not both, we found that the Bayesian approach is more robust to outlier samples within a single dataset than the frequentist approach. For example, *SLC6A4* was significant by the frequentist approach (FDR<4%), but not by the Bayesian approach (p = 0.24). Comparison of effect sizes from both approaches showed that for GSE64913, the frequentist approach estimated statistically significant non-zero effect size, whereas the Bayesian approach estimated non-significant effect size (**Fig 2E**). Further analysis showed that although 59 out of 70 samples in GSE64913 had identical expression values for *SLC6A4*, irrespective of estimate of statistically significant effect size by the frequentist approach was driven by only 2 out of 70 samples (**Fig 2F**). In contrast, the Bayesian approach correctly estimated near-zero effect size due to its reliance on parameter estimation and sampling, and was not confounded by a small number of outliers. This observation demonstrated that the Bayesian approach reduced false positives.

In contrast, another gene, *FANCI*, was statistically significant by the Bayesian approach (p = 0.01), but not by the frequentist approach (FDR = 19%). Although both approaches estimated the effect size for *FANCI* in the same direction, the difference in statistical significance was due to the difference in estimated effect size in a single dataset (GSE18965) (**Fig 2E**). The frequentist approach estimated negative effect size for *FANCI* in GSE18965, which was driven by a single healthy control sample (**Fig 2F**), which in turn led to higher between-study heterogeneity for the gene and the summary effect size for the gene being statistically insignificant. In contrast, the Bayesian approach was not confounded by a single sample, correctly estimated its effect size as positive in GSE18965, and identified it as being statistically significant overall. Collectively, these results show that the Bayesian meta-analysis approach is robust to outliers, which in turn reduces false positives and false negatives.

## Comparing Bayesian and frequentist meta-analysis methods

A key difference between the frequentist and Bayesian meta-analysis approaches is how between-study heterogeneity affects estimates of summary effect size and its statistical significance. To investigate the effect of between-study heterogeneity, we simulated data across 5 independent studies (**Methods**) such that either the effect size, its variance, or both changed for one study (**Fig 3**). Increasing the effect size for one study without changing its variance estimate, which simulated adding a study with high certainty of a strong positive effect, statistical significance reduced for the frequentist approach (i.e., higher FDR), but increased for the Bayesian approach (i.e., lower probability), although the between-study heterogeneity increased for both approaches (**Fig 3B**). When we increased only variance or both effect size and variance, the summary heterogeneity and statistical significance decreased for both approaches (**Fig 3C and 3D**). These results, combined with robustness of the Bayesian approach to outliers within a dataset, are desired characteristics for identifying generalizable signal across heterogeneous datasets.

## Comparing the predictive performance of Bayesian and frequentist meta-analysis

Given that the Bayesian approach is robust to outlier samples within a study and better estimates between-study heterogeneity, we investigated whether it would identify DEGs that are more generalizable to unseen data than the frequentist approach. To investigate this, we applied both meta-analysis approaches to transcriptome profiles from patients with cardiomyopathy (14 datasets, 1039 samples) [13–23] or tuberculosis (27 datasets, 3069 samples) [24–45]. For both diseases, we identified most differentially expressed genes by successively increasing the number of datasets and compared their discriminatory power in unused

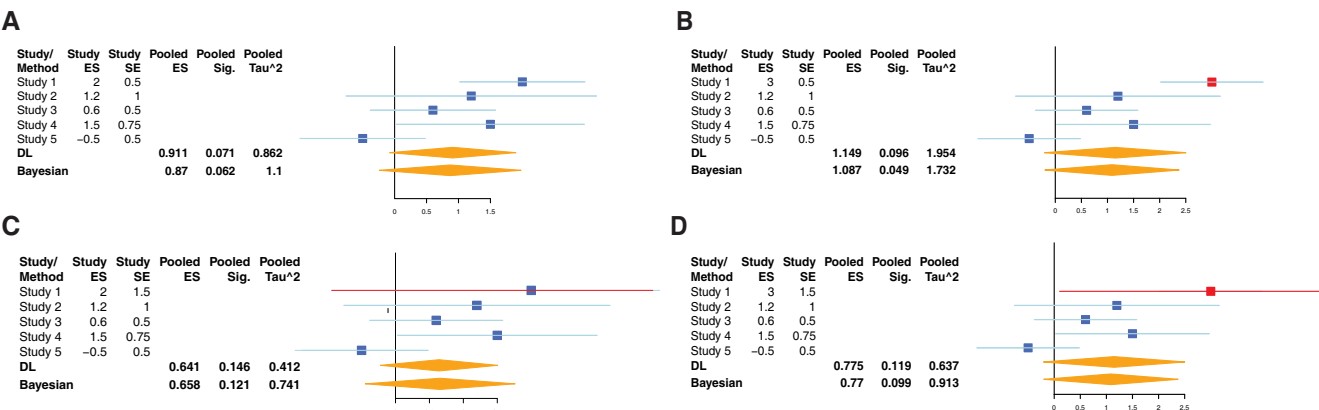

**Fig 3. Simulated Meta-Analysis.** A) Using simulated data in five studies, we explored the effect of changing the effect size and variance of a single study on each model's pooled effect size, heterogeneity ($\tau^2$) and significance. Here, we show the baseline meta-analysis. We picked a scenario where both Bayesian and frequentist are borderline significant. B) Increasing effect size. We increased the effect size of a single study–creating high certainty of a strong positive effect of that study. The Bayesian model became more significant, whereas the frequentist model became less significant. Both increased in effect size and $\tau^2$. C) Increasing variability. We increased the variability of a single study. We found that both methods behave similarly–with effect size, significance and $\tau^2$ decreasing. D) Increasing variability and effect size. We increased both the variability and effect size of a single study. We found that both methods behave similarly–with effect size, significance and $\tau^2$ decreasing.

datasets (**Fig 4A** and **Methods**). For both diseases, we randomly selected *N* datasets 100 times and applied both meta-analysis approaches to each set of randomly selected *N* datasets. For each iteration, we selected DEGs with absolute summary effect size > 0.6 and used the top 10 genes with the lowest FDR or Bayesian probability. We used difference of geometric means of over- and under-expressed genes as a classifier in unseen datasets to distinguish cases from healthy controls. We chose difference of geometric mean as a classification model because such a classifier has been repeatedly demonstrated to be generalizable and has been translated in a point-of-care test [5,6,46,47]. When we varied the effect size threshold (0.4 to 1.1) and the number of selected genes (10 to 200), while keeping the number of datasets used for analysis constant at 4, the genes selected using the Bayesian approach consistently led to higher AUC (**S2 Fig**).

First, we compared area under the receiver operating characteristic (AUROC) curves in the unused datasets in each iteration as a proxy for identifying generalizable gene signatures for both meta-analysis approaches. For both diseases, irrespective of the number of datasets used, the DEGs identified using the Bayesian approach had consistently higher AUROC in unseen datasets than using the frequentist approach. When using 2 out of 27 datasets to identify DEGs for tuberculosis, there was no significant difference in AUROC between the two approaches, which suggests that *N* = 2 does not represent the heterogeneity across the other 27 datasets. Interestingly, for both diseases, the median AUROCs for the Bayesian approach using 3 datasets was always equal or greater than the median AUROCs for the frequentist approach using substantially larger number of datasets (9 datasets for cardiomyopathy, 14 datasets for tuberculosis). This result demonstrated that the DEGs identified using the Bayesian approach are more generalizable to previously unseen data than those identified using the frequentist approach. Further, it also suggests that using as few as 3 datasets may be sufficient to identify robust gene signatures of a disease using the Bayesian meta-analysis approach.

Next, we investigated between-study heterogeneity, $\tau^2$, and effect sizes for the DEGs identified by the two approaches and whether those differences explained more generalizability for the Bayesian approach. For both diseases, median $\tau^2$ decreased with the increasing number of datasets (**Fig 4C**). The DEGs identified by the Bayesian approach had significantly lower $\tau^2$

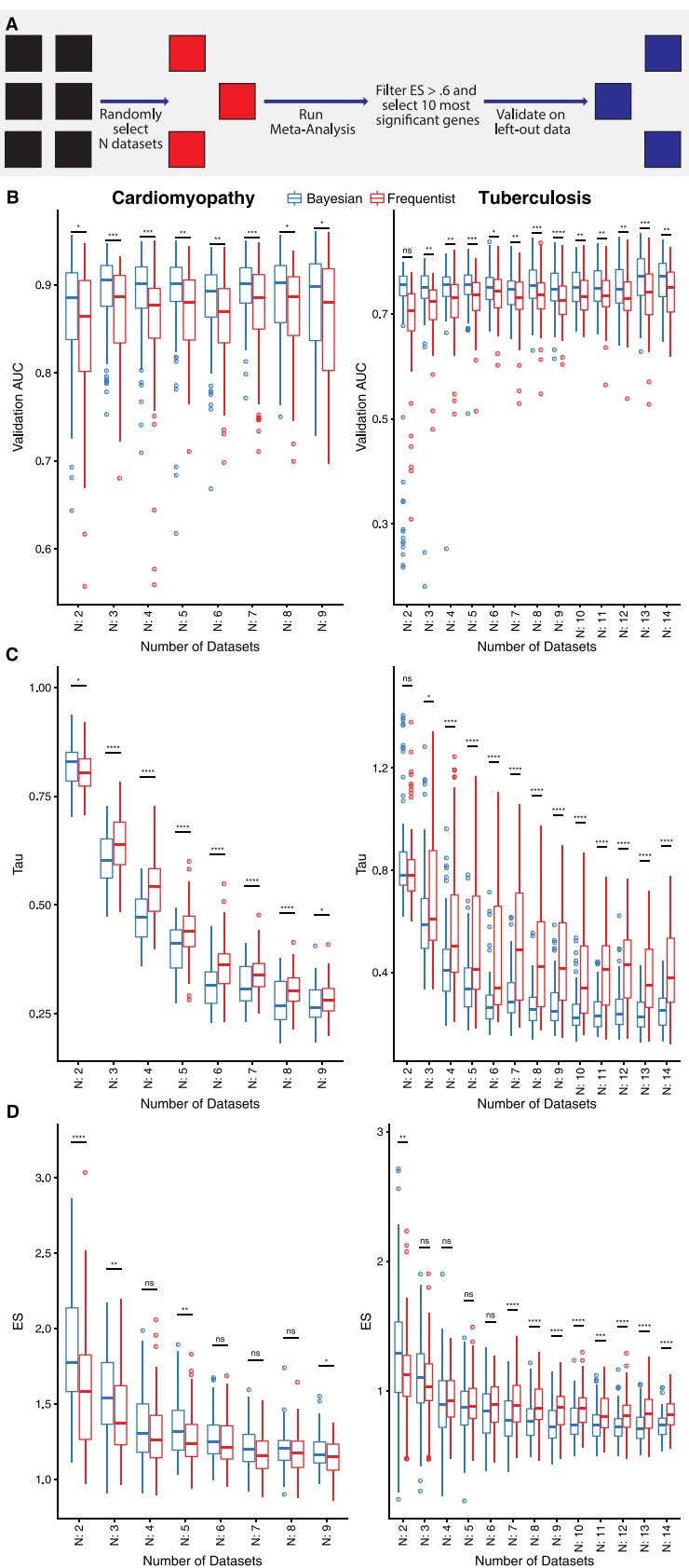

**Fig 4. Comparing gene signatures from frequentist and Bayesian meta-analysis.** A) Selection method—for both diseases, we randomly selected *N* datasets 100 times. For each iteration, we selected DEGs with absolute pooled effect size > 0.6 and selected 10 genes with the smallest FDR or Bayesian probability. The signature is then created using a geometric mean signature score and validated on completely left-out datasets. B) Average AUC–the Bayesian method picked gene signatures that are more generalizable—achieving higher AUCs for both diseases on left-out datasets. The Bayesian method also requires less datasets to achieve higher classifier performance. C) Average τ–we find that the between study heterogeneity (τ) is significantly lower for the gene signatures selected by the Bayesian method compared to the frequentist, suggesting the Bayesian method is selecting genes that have lower heterogeneity. Due to the frequentist method calculating τ as 0 for a large percentage of genes, Bayesian tau is displayed for all genes. D) Mean pooled effect size–we do not see consistent trends in the average effect sizes of the gene signatures, except for the tuberculosis analysis where the frequentist method tended to pick higher effect size genes with larger Ns. **** < 0.0001, *** < 0.001, ** < 0.01, * < 0.05, ns: > 0.05.

than those identified by the frequentist approach. Interestingly, for the frequentist approach as the number of datasets increased, variability in $\tau^2$ increased for tuberculosis analysis but not cardiomyopathy analysis. However, there was no consistent difference in the summary effect sizes of the genes selected by either approach. For example, for cardiomyopathy datasets, the difference in summary effect sizes of DEGs were mostly non-significant, whereas for tuberculosis datasets, there was no statistically significant difference in summary effect sizes, when using a smaller number of datasets ($N\leq6$), but the frequentist approach selected genes with significantly higher summary effect sizes for a larger number of datasets ($N>6$). Interestingly, for tuberculosis analysis, higher summary effect size and variability in $\tau^2$ with the increased number of datasets suggest that the DEGs may be affected by outlier datasets, which in turn reduces their generalizability to unseen datasets with reduced AUROCs.

To further investigate the differences between gene sets created using Bayesian or frequentist meta-analysis, we performed an overall meta-analysis with all of the Asthma, Cardiomyopathy and Tuberculosis datasets. Calculating the Jaccard similarity of the gene sets, we found that for the smallest meta-analysis, asthma, the Jaccard similarity between the models was low for the top 500 genes, only reaching 0.3. For the larger meta-analyses, Cardiomyopathy and Tuberculosis, we found that the Jaccard similarity increased up to 0.5 (**S3 Fig**). In each case, the Jaccard similarity increased until reaching a plateau. We also compared the gene sets using pathway analysis using ReactomePA [48] (**S4 Fig**). The pathways represented were similar in both analysis types for all three analyses, which suggested that although Bayesian meta-analysis identified more generalizable genes that have higher discriminatory power, it is still identifying the same biological pathways as frequentist meta-analysis.

Collectively, our results show that compared to the frequentist approach, the Bayesian approach for meta-analysis identifies genes with lower between-study heterogeneity and comparable summary effect sizes, and is robust to outlier samples, which in turn leads to more generalizable classifier for unseen datasets. Our results also suggest that the Bayesian approach requires lower number of datasets to identify generalizable DEGs compared to the frequentist approach.

## Discussion

We performed three gene expression meta-analyses to compare the Bayesian and frequentist meta-analysis approaches. Using dozens of publicly available gene expression studies, we found that Bayesian approach tends to identify differentially expressed genes that have lower between-dataset heterogeneity and higher discriminatory power, which leads to more generalizable classifiers. Importantly, we found that the Bayesian approach consistently required lower number of datasets than the frequentist approach.

Several factors contribute to drive these effects for the Bayesian meta-analysis approach. First, our analysis showed that the Bayesian approach is resistant to outliers due to the *t*-

distribution underlying the estimation of the effect size distribution per dataset. Second, the Bayesian approach uses probabilistic distributions to represent effect size in each dataset as opposed to confidence intervals that the frequentist approach relies on. When the number of datasets used for meta-analysis is small and the within-dataset variability is higher, confidence intervals tend to be wider, leading the frequentist approach to estimate the between-study heterogeneity as zero or very low for a large number of genes the large within dataset heterogeneity. In contrast, the use of probabilistic distribution leads to conservative estimates of between-study heterogeneity. For biomarker discovery, this is preferable, for we seek to find the biomarkers that have the smallest between-study heterogeneity across all datasets in our analysis. Finally, the p-values in the frequentist approach represent the probability of observing data under a hypothesis of no effect and must be multiple hypothesis adjusted, whereas Bayesian probability represents the posterior belief of the difference between groups and require no multiple hypothesis adjustment [49].

Although the Bayesian meta-analysis tends to perform better than the frequentist method at finding consistently differentially expressed genes across studies with low heterogeneity, one area in which it would be less advantageous is the unsupervised identification of subgroups within patient populations. In this case, we would want to select for genes that have high effect sizes when compared to controls, but also have heterogeneity to separate between cases. For this task, the frequentist method would likely be more effective than the Bayesian method described in this study. However, priors and probability calculations can be adjusted, providing Bayesian meta-analysis the flexibility to succeed in many different scenarios. A limitation of our study is that we used minimally informative priors for all estimation in order to produce the most accurate estimates of effect size and heterogeneity. However, depending on the context, these priors could be changed. For example, in the case of finding biomarkers for diagnostic use, the prior for $\tau^2$ could be changed from a uniform to a monotonically increasing function. This would in turn create a form of regularization, pushing $\tau^2$ to be estimated as larger and increasing the effect of heterogeneity on Bayesian probability estimate. For subgroup identification and clustering of patients, one could similarly adjust priors and probability estimates to select genes that have high effect size and moderate-to-high within and/or between study heterogeneity. This shows the true flexibility and potential adaptability of the Bayesian framework for different uses.

## Methods

### Dataset selection

We used publicly available transcriptome data from the NCBI GEO for three diseases: (1) 223 samples across 4 datasets from healthy controls and patients with asthma (**S1 Table**) [10–12], (2) 1039 samples across 14 datasets from healthy controls and patients with cardiomyopathy (**S2 Table**) [13–23], and (3) 3069 samples across 27 datasets from healthy controls and patients with tuberculosis (TB) (**S3 Table**) [24–45]. Each dataset was appropriately normalized and log2 transformed, if not already in log scale. We removed genes that were not present in at least half of the datasets for a given disease.

### Frequentist meta-analysis

We used the frequentist meta-analysis implemented in MetaIntegrator, which uses random effects inverse variant model, for comparison with the Bayesian method [50]. Briefly, MetaIntegrator computes a Hedge's *g* as an effect size for each gene in each dataset. The effect sizes are combined using random effects inverse variance model the DerSimonian-Laird method, and the corresponding p-value is estimated using a standard normal distribution, which is

corrected for multiple hypotheses testing using the Benjamini-Hochberg FDR adjustment [51]. Following these calculations, the top genes are selected by using FDR and effect size thresholds. We used difference between geometric mean of over-expressed genes to that of under-expressed genes as a classifier because it has been repeatedly shown to be more generalizable across datasets [52] and has also been translated in a point-of-care test [5,6,46,47].

## Bayesian meta-analysis–dataset effect size calculation

The first step of the Bayesian meta-analysis pipeline involves creating an effect size distribution for each case and control for each gene in each dataset (**Fig 1**). We used the BEST [8] framework with default parameters and priors for this purpose. The BEST framework estimates the posterior distribution of effect size between cases and controls for each gene in each dataset by assuming that the data is independently distributed and comes from a $t$ distribution with different mean ($\mu$) and standard deviation ($\sigma$) parameters for each group. Then, we combine the distributions from independent studies using a gaussian hierarchical model, estimating both the pooled effect size and between-study heterogeneity in the process. Overall normality parameter ($\nu$) that denotes the size of tails of the $t$ distribution and the level of normality. Overall, the BEST framework estimates 5 parameters: $\mu_1$, $\mu_2$, $\sigma_1$, $\sigma_2$ and $\nu$ using minimally informative priors. $\mu_1$ and $\mu_2$ are the population means of cases and controls and are parameterized with a wide normal prior with a large standard deviation. $\sigma_1$ and $\sigma_2$ are the population standard deviations of cases and controls and are parameterized with a broad uniform. The normality parameter, $\nu$, has a broad, shifted exponential prior. Following the parameter estimation, we calculated the effect size as a standardized mean difference:

$$\frac{(\mu_1 - \mu_2)}{\sqrt{(\sigma_1 + \sigma_2)/2}}$$

For this study, we ran all individual datasets with 2000 steps with 400 for model burn-in with 3 chains to ensure convergence, which is then calculated using $\hat{r}$ [53]. We removed any genes with an $\hat{r}$ greater than 1.1 from the dataset.

## Bayesian meta-analysis–pooling step

Following the dataset effect size distribution estimation, a pooling step is performed to estimate an overall pooled effect size using a hierarchical model. The effect size distribution from each dataset for each gene is assumed to be normally distributed with mean $\mu_i$ and $\sigma_i$. To calculate the pooled effect size, we use each of the calculated dataset effect size distributions and assume that they are sampled from an overall, pooled normal distribution represented as $Normal(\mu_{pooled}, \tau^2)$. Both $\mu$ and $\tau$ are parameterized with minimally informative priors: $\mu_{pooled}$ as $Normal(0, 3)$ and $\tau$ as $Uniform(0, 2)$ and parameters are estimated using Gibbs sampling [54]. We chose priors for effect size and between-study heterogeneity using 122 previous gene expression meta-analyses (**S1A** and **S1B Fig**) [50]. Sensitivity analysis of the priors found that when we varied the parameters of the Normal priors for effect size, the Bayesian probabilities for differential expression remained concordant (**S1C** and **S1D Fig**). For between-study heterogeneity, we found that using Uniform(0,1) as prior had lower Bayesian probability estimates (**S1E Fig**), whereas using Uniform(0,2) or Uniform(0,3) as prior had highly concordant Bayesian probability estimates (**S1F Fig**). Hence, the final gene rankings and posterior probabilities did not change by widening the priors further.

For this study, all pooling steps were run with 5000 steps with 1000 for burn-in with 3 chains. The convergence parameter ($\hat{r}$) is calculated using the chains and any genes with an $\hat{r}$

greater than 1.1 are removed. Using the hierarchical model structure, once the pooled distribution is estimated, we adjusted the individual dataset effect size distributions based on the summary distribution, akin to a random effects model in frequentist meta-analysis. To calculate statistical significance, we calculate the probability that a certain gene is upregulated or downregulated, calculating $Pr(\mu_{pooled} < 0)$ or $Pr(\mu_{pooled} > 0)$, respectively. This is done with a standard cumulative density function for a normal distribution.

### Simulated data for comparison of frequentist and Bayesian meta-analysis

Random study data was used to compare the Bayesian and frequentist methods. The frequentist random-effects meta-analysis was run using the metagen function from the meta package [55] in the R using default parameters. The Bayesian model was run as with the parameters described above, with 50000 steps with 10000 for burn-in.

### Comparison of AUCs, effect sizes, and heterogeneity between frequentist and Bayesian meta-analysis

To compare the methods, we used the Tuberculosis and Cardiomyopathy cohorts, as defined above. We removed genes that were not present in at least half of the datasets. For both diseases, we randomly selected *N* datasets 100 times (or 91 times in the case of Cardiomyopathy: N = 2) and applied both meta-analysis approaches Bayesian (bayesMetaIntegrator) and frequentist (MetaIntegrator) to each set of randomly selected *N* datasets. For each iteration, we filtered to differentially expressed genes with absolute summary effect size > 0.6 and used the top 10 genes with the smallest FDR or Bayesian probability. We used difference of geometric means of over- and under-expressed genes as a classifier in unseen datasets to distinguish cases from healthy controls, as defined below. For the tau comparison, we report the Bayesian estimate of tau for each gene due to frequentist meta-analysis reporting large numbers of genes as 0. For pathway analysis, the ReactomePA package [48] was used to perform pathway analysis on all three diseases, using the most significant genes for both methods– 1000 genes for Cardiomyopathy and Tuberculosis and 500 genes for Asthma. A p-value cutoff of 0.2 was used for Cardiomyopathy, 0.2 for Asthma and 0.05 for Tuberculosis.

## Supporting information

**S1 Fig.** A) Using 122 previous gene expression meta-analyses we observed the pooled effect size and tau to determine our initial priors. Our priors, picked to be minimally informative, are shown in red. B) Sensitivity analysis of effect size prior using Asthma data. C) Sensitivity analysis of tau prior using Asthma data.
(EPS)

**S2 Fig.** A) Using N = 4 datasets and top 10 statistically significant genes, we examined the effect of effect size thresholds using the AUC performance on all other left-out datasets. We find that the Bayesian model consistently outperforms the frequentist at effect sizes < 1. B) Using N = 4 datasets and effect size threshold of .6, we examined how the number of genes in a signature affect performance. We find that no matter how many genes were used in the signature, from 10–200, the Bayesian model consistently outperforms the frequentist.
(EPS)

**S3 Fig. Jaccard similarity of the top N genes, by statistical significance, between the Bayesian and frequentist meta-analysis methods, which is generally low.**
(EPS)

**S4 Fig. Pathway analysis using Reactome PA—using the most significant genes for both methods– 1000 genes for Cardiomyopathy and Tuberculosis and 500 genes for Asthma.** A p-value cutoff of .2 was used for Cardiomyopathy, .2 for Asthma and .05 for Tuberculosis. BMA_only denotes genes only significant in Bayesian, but not frequentist. FMA_only denotes genes only significant in frequentist, but not Bayesian.
(EPS)

**S1 Table. Datasets used for meta-analysis of asthma.**
(XLSX)

**S2 Table. Datasets used for meta-analysis of cardiomyopathy.**
(XLSX)

**S3 Table. Datasets used for meta-analysis of tuberculosis.**
(XLSX)

## Acknowledgments

We would like to acknowledge Michele Donato and Ian Lee, for their contributions and work testing the method and package. We would like to thank the researchers that have contributed the datasets used within this study.

## Author Contributions

**Conceptualization:** Laurynas Kalesinskas, Purvesh Khatri.

**Data curation:** Laurynas Kalesinskas, Sanjana Gupta.

**Formal analysis:** Laurynas Kalesinskas, Purvesh Khatri.

**Funding acquisition:** Purvesh Khatri.

**Investigation:** Laurynas Kalesinskas, Sanjana Gupta, Purvesh Khatri.

**Methodology:** Laurynas Kalesinskas.

**Project administration:** Purvesh Khatri.

**Resources:** Purvesh Khatri.

**Software:** Laurynas Kalesinskas, Sanjana Gupta.

**Supervision:** Purvesh Khatri.

**Validation:** Laurynas Kalesinskas.

**Visualization:** Laurynas Kalesinskas.

**Writing – original draft:** Laurynas Kalesinskas, Purvesh Khatri.

**Writing – review & editing:** Laurynas Kalesinskas, Purvesh Khatri.

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
