## [Decision Letter · Decision Letter 0]

27 Feb 2022

Dear Dr. Khatri,

Thank you very much for submitting your manuscript "Increasing reproducibility, robustness, and generalizability of biomarker selection from meta-analysis using Bayesian methodology" for consideration at PLOS Computational Biology.

As with all papers reviewed by the journal, your manuscript was reviewed by members of the editorial board and by several independent reviewers. In light of the reviews (below this email), we would like to invite the resubmission of a significantly-revised version that takes into account the reviewers' comments.

Please address the concerns and comments raised by the reviewers; specifically, please clarify the choice of prior distributions and hyperparameters and assess their effects on the overall performance.

We cannot make any decision about publication until we have seen the revised manuscript and your response to the reviewers' comments. Your revised manuscript is also likely to be sent to reviewers for further evaluation.

Sincerely,

Yuchao Jiang, Ph.D.

Guest Editor

PLOS Computational Biology

Jian Ma

Deputy Editor

PLOS Computational Biology

Please address the concerns and comments raised by the reviewers; specifically, please clarify the choice of prior distributions and hyperparameters and assess their effects on the overall performance.

Reviewer's Responses to Questions

**Comments to the Authors:**

Reviewer #1: Enclosed is a review of Kalesinskas et al’s manuscript

“Increasing reproducibility, robustness, and generalizability of biomarker selection from meta2 analysis using Bayesian methodology.”

In this paper, the authors describe a Bayesian meta-analysis approach that, in comparison to traditional frequentist approaches, is robust to outliers, increase the informativeness of between-study heterogeneity estimates, and reduces false positives and negatives. Overall, the paper is well-written and clearly motivated and explained. There are few comments I have that I detail section-by-section below. In summary, I think there may be room for added clarity for some figures.

INTRODUCTION

1. I commend the authors for clearly articulating this problem both from the statistical and biological perspectives.

2. In the last sentence of the Introduction, I suggest the authors name their method and R package.

RESULTS

1. The forest plots (like in Figure 2E) are hard to read and there is limited information about them in the figure captions. For example, what do the differently sized blue boxes mean?

2. I don’t quite see how effect sizes for FANCI are in opposite directions. The text seems to suggest the effect sizes are in opposite directions for the meta-analysis, but Figure 2E only shows different directions for one dataset.

3. The asterisks in Figure 4 are not defined. Are these P-value adjusted for multiple comparisons?

METHODS

1. How were the minimally informative prior hyperparameters elicited here? I.e., the Normal(0,3) and Uniform(0,2)?

Reviewer #2: In this study, the authors propose a Bayesian methodology for gene expression meta-analysis and show strong evidence that it outperforms standard frequentist approaches, and is, in particular, much less sensitive outlying data points and more sensitive to between-dataset heterogeneity. Overall, the study is well designed and the paper is well organized and written. I have no major concerns.

While I have no reservations about the final results, I think it would be helpful to include sensitivity analyses of the Bayesian prior parameters to assure that they are indeed non-informative, i.e. that the final gene rankings and posterior probabilities do not change by widening the prior further. I fully expect this to be the case.

Reviewer #3: The review is uploaded as an attachment.

**Have the authors made all data and (if applicable) computational code underlying the findings in their manuscript fully available?**

Reviewer #1: Yes

Reviewer #2: Yes

Reviewer #3: **No: **Authors provided a Github repo for the software (https://github.com/Khatri-Lab/bayesMetaIntegrator), but the codes and data to reproduce the analyses in this manuscript have not been provided.

PLOS authors have the option to publish the peer review history of their article (what does this mean?). If published, this will include your full peer review and any attached files.

Reviewer #1: No

Reviewer #2: No

Reviewer #3: No
---

## [Decision Letter · Decision Letter 1]

29 May 2022

Dear Dr. Khatri,

We are pleased to inform you that your manuscript 'Increasing reproducibility, robustness, and generalizability of biomarker selection from meta-analysis using Bayesian methodology' has been provisionally accepted for publication in PLOS Computational Biology.

Best regards,

Yuchao Jiang, Ph.D.

Guest Editor

PLOS Computational Biology

Jian Ma

Deputy Editor

PLOS Computational Biology

Reviewer's Responses to Questions

**Comments to the Authors:**

Reviewer #1: My comments have been addressed; I have no further comment. I congratulate the authors on their work and recommend publication!

Reviewer #2: The authors have addressed all of my comments.

Reviewer #3: The authors have answered all my questions. I don't have further questions.

**Have the authors made all data and (if applicable) computational code underlying the findings in their manuscript fully available?**

Reviewer #1: Yes

Reviewer #2: None

Reviewer #3: Yes

PLOS authors have the option to publish the peer review history of their article (what does this mean?). If published, this will include your full peer review and any attached files.

Reviewer #1: **Yes: **Arjun Bhattacharya

Reviewer #2: No

Reviewer #3: No

---

## [Editor Report · Acceptance letter]

22 Jun 2022

PCOMPBIOL-D-22-00035R1 

Increasing reproducibility, robustness, and generalizability of biomarker selection from meta-analysis using Bayesian methodology

Dear Dr Khatri,

I am pleased to inform you that your manuscript has been formally accepted for publication in PLOS Computational Biology. Your manuscript is now with our production department and you will be notified of the publication date in due course.

With kind regards,

Zsofia Freund
